# Pyridoxamine Limits Cardiac Dysfunction in a Rat Model of Doxorubicin-Induced Cardiotoxicity

**DOI:** 10.3390/antiox13010112

**Published:** 2024-01-17

**Authors:** Sibren Haesen, Manon Marie Jager, Aline Brillouet, Iris de Laat, Lotte Vastmans, Eline Verghote, Anouk Delaet, Sarah D’Haese, Ibrahim Hamad, Markus Kleinewietfeld, Jeroen Mebis, Wilfried Mullens, Ivo Lambrichts, Esther Wolfs, Dorien Deluyker, Virginie Bito

**Affiliations:** 1UHasselt, Faculty of Medicine and Life Sciences, Biomedical Research Institute (BIOMED), Agoralaan, 3590 Diepenbeek, Belgium; sibren.haesen@uhasselt.be (S.H.); manonmarie.jager@uliege.be (M.M.J.); alinebrillouet@hotmail.com (A.B.); iris.delaat2000@gmail.com (I.d.L.); lotte.vastmans@student.uhasselt.be (L.V.); everghote@hotmail.be (E.V.); anouk.delaet@student.uhasselt.be (A.D.); sarah.dhaese@uhasselt.be (S.D.); ibrahim.hamad@uhasselt.vib.be (I.H.); markus.kleinewietfeld@uhasselt.vib.be (M.K.); jeroen.mebis@uhasselt.be (J.M.); wilfried.mullens@uhasselt.be (W.M.); ivo.lambrichts@uhasselt.be (I.L.); esther.wolfs@uhasselt.be (E.W.); dorien.deluyker@uhasselt.be (D.D.); 2Cardiovascular Research Institute Maastricht (CARIM), School for Cardiovascular Diseases, University of Maastricht, Universiteitssingel 50, 6229 ER Maastricht, The Netherlands; 3VIB Laboratory of Translational Immunomodulation, VIB Center for Inflammation Research (IRC) Hasselt University, 3590 Diepenbeek, Belgium; 4Department of Medical Oncology, Jessa Hospital, Stadsomvaart 11, 3500 Hasselt, Belgium; 5Department of Cardiology, Ziekenhuis Oost Limburg, Schiepse Bos 6, 3600 Genk, Belgium

**Keywords:** anthracyclines, cardiotoxicity, pyridoxamine, cardioprotection, preclinical study, inflammation, redox biology, mitochondria

## Abstract

The use of doxorubicin (DOX) chemotherapy is restricted due to dose-dependent cardiotoxicity. Pyridoxamine (PM) is a vitamin B6 derivative with favorable effects on diverse cardiovascular diseases, suggesting a cardioprotective effect on DOX-induced cardiotoxicity. The cardioprotective nature of PM was investigated in a rat model of DOX-induced cardiotoxicity. Six-week-old female Sprague Dawley rats were treated intravenously with 2 mg/kg DOX or saline (CTRL) weekly for eight weeks. Two other groups received PM via the drinking water next to DOX (DOX+PM) or saline (CTRL+PM). Echocardiography, strain analysis, and hemodynamic measurements were performed to evaluate cardiac function. Fibrotic remodeling, myocardial inflammation, oxidative stress, apoptosis, and ferroptosis were evaluated by various in vitro techniques. PM significantly attenuated DOX-induced left ventricular (LV) dilated cardiomyopathy and limited TGF-β1-related LV fibrotic remodeling and macrophage-driven myocardial inflammation. PM protected against DOX-induced ferroptosis, as evidenced by restored DOX-induced disturbance of redox balance, improved cytosolic and mitochondrial iron regulation, and reduced mitochondrial damage at the gene level. In conclusion, PM attenuated the development of cardiac damage after DOX treatment by reducing myocardial fibrosis, inflammation, and mitochondrial damage and by restoring redox and iron regulation at the gene level, suggesting that PM may be a novel cardioprotective strategy for DOX-induced cardiomyopathy.

## 1. Introduction

According to the World Health Organization, cardiovascular disease (CVD) and cancer are the leading mortality causes, with respectively, 17.9 million and nearly 10 million yearly deaths worldwide [1,2]. Over the last decades, cancer prognosis has improved tremendously due to earlier detection and treatment advances, resulting in a better cancer prognosis [3]. The anthracycline drug doxorubicin (DOX) is a highly effective cancer drug that is prominent in treating lymphoma, sarcoma, leukemia, and breast cancer. However, clinical use is limited due to its dose-dependent cardiotoxicity, characterized by an enlarged left ventricle (LV), weakened heart muscle, and a decline in LV ejection fraction (LVEF) [4]. Almost 40% of anthracycline-treated cancer patients experience mild, moderate, or severe cardiotoxicity, and CVD accounts for 11% of mortality among breast cancer patients [5,6]. The pathogenesis of DOX-induced cardiotoxicity is multifactorial, involving fibrotic and inflammatory tissue remodeling, disturbed redox balance leading to oxidative stress, and mitochondrial damage in cardiomyocytes [7]. Growing evidence indicates that DOX also disrupts cellular and mitochondrial iron metabolism, leading to iron overload in cardiomyocytes and iron-dependent cell death called ferroptosis [8,9,10,11]. In addition, DOX hampers DNA transcription and replication by inhibiting DNA topoisomerase II beta (TOP2β) activity, resulting in double-stranded DNA breaks and cardiomyocyte death [12].

The 2022 ESC guidelines on cardio-oncology acknowledge that there is limited evidence (level C) and weak recommendation (class IIa/b) for the routine use of cardiovascular drugs, including neurohormonal therapies and β-blockers, in cancer patients receiving anthracycline chemotherapy [13]. Although multiple clinical trials showed beneficial effects [14,15], a recent meta-analysis concluded that the clinical significance of small short-term absolute changes in LVEF with neurohormonal therapy is uncertain, treatment effects on clinical outcomes are lacking, and more extensive clinical trials are needed to confirm findings [16]. Moreover, clinical practice often does not allow the initiation of these agents in frail anorectic cancer patients with volume shifts and low blood pressure. The use of DRZ, an iron-chelating agent and inhibitor of DOX-TOP2β complex formation and the only FDA-approved drug to prevent anthracycline-induced cardiotoxicity, is only recommended for patients at high risk for cardiotoxicity or for those who received high cumulative anthracycline doses [13,17,18]. In addition, its clinical use has been restricted due to safety issues and suspicions about potential interference with the antitumor activity of DOX and the risk of second primary malignancies [19]. Consequently, cardioprotection in DOX-treated cancer patients is suboptimal. Previous animal studies have shown that the vitamin B6 derivative pyridoxamine (PM) limits the progression of CVDs such as myocardial ischemia and diabetes-associated atherosclerosis [20,21] without inducing side effects. The potential cardioprotective effects of PM are mainly related to its ability to chelate metal ions and reduce reactive oxygen species (ROS) formation and inflammation [22,23].

This study aims to investigate whether and how PM is cardioprotective against DOX-induced cardiotoxicity. In this study, we confirmed that PM attenuates the development of DOX-induced LV dilated cardiomyopathy by reducing myocardial fibrosis, inflammation, and mitochondrial damage and by limiting the disturbance of cellular and mitochondrial iron regulation.

## 2. Materials and Methods

### 2.1. Animal Model

All animal experiments follow the EU Directive 2010/63/EU for animal testing and are approved by the local ethical committee (Ethical Commission for Animal Experimentation, UHasselt, Diepenbeek, Belgium, ID 201942 and ID 202154). Animals were group-housed and fed a standard pellet diet with water available ad libitum. Adult six-week-old female Sprague Dawley rats (Janvier Labs, Le Genest-Saint-Isle, France) were randomly allocated into either the DOX group (2 mg/kg, once a week for eight weeks, intravenously injected, N = 15, Accord Healthcare B.V., Utrecht, The Netherlands) or the control group (CTRL, an equal volume of 0.9% saline injections, once a week for eight weeks, intravenously injected, N = 14). The protocol and cumulative dose for the induction of chronic DOX cardiotoxicity are based on current guidelines on the use of preclinical models in cardio-oncology [24], with both the intravenous injection route and the eight-week study duration adding up to the translatability of our animal model [24,25,26]. The cumulative DOX dose of 16 mg/kg equals the clinically relevant dose of 592 mg/m² [27]. Female rats were used in this study because the female sex is considered a risk factor for the development of anthracycline cardiotoxicity [28], to compare with later studies using a model of combined breast cancer and DOX-induced cardiomyopathy, and because of its underrepresentation in preclinical studies. Two other groups received pyridoxamine dihydrochloride (PM, 1 g/L dissolved in the drinking water, sc-219673, Santa Cruz Biotechnology, Inc., Heidelberg, Germany) continuously for eight weeks in addition to DOX (DOX+PM, N = 18) or saline (CTRL+PM, N = 14), starting from the first injection.

### 2.2. Study Design

This study design and animal model are illustrated in Appendix A. Body weight was measured weekly. Blood samples were obtained at baseline and week 8 to quantify plasma brain natriuretic peptide (BNP) levels with ELISA. Rats were anesthetized with 2% isoflurane supplemented with oxygen (0.8 L/min), induced via inhalation, to perform echocardiography and hemodynamic measurements. Echocardiography was performed at baseline and week 8 to assess conventional and non-conventional cardiac parameters. Invasive LV and right ventricular (RV) hemodynamic measurements were performed at the end of week 9, and immediately thereafter, the rats were injected with heparin (1000 u/kg intraperitoneal) and were euthanized with an overdose of sodium pentobarbital (150 mg/kg intraperitoneal, Dolethal, Vetoquinol, Aartselaar, Belgium). The heart and lungs were isolated and weighed. Lung weight was measured before and after dehydration at 60 °C for five days to calculate the wet-to-dry weight ratio. All analyses were performed by two independent and blinded researchers.

### 2.3. Echocardiography

#### 2.3.1. Conventional Echocardiography and 4D Imaging

Transthoracic echocardiography of the left ventricle (LV) was performed in a supine position under 2% isoflurane anesthesia supplemented with oxygen with a Vevo^®^ 3100 high-resolution imaging system and a 21 MHz MX250 transducer (FUJIFILM VisualSonics, Inc., Amsterdam, The Netherlands) as described before [29] and according to the standards proposed by Zacchigna et al. 2021 [30]. Before imaging, the thorax was shaved, and depilatory cream was applied to minimize artifacts. While scanning, vital physiological parameters, including heart rate (HR), respiratory rate, body temperature, and ECG signals, were monitored noninvasively using the accompanying Vevo Imaging Station. From parasternal long-axis views in B-mode (EKV), LV longitudinal fractional shortening (LVFS) was measured by tracing the LV endocardial border in end-systole and end-diastole. LV anterior and posterior wall thickness in diastole (LVAW_d_ and LVPW_d_) were measured from parasternal short-axis views in M-mode, for which measurements from at least three heartbeats were averaged. Four-dimensional (4D) views were acquired to measure LV end-systolic and end-diastolic LV volumes (LVESV and LVEDV), LV stroke volume (LVSV), LV ejection fraction (LVEF), and LV cardiac output (LVCO). LVSV and LVCO were normalized to body surface area and expressed as the LVSV index and LV cardiac index. Apical four-chamber views were acquired in B-mode to assess mitral flow with Pulsed-Wave (PW) Doppler mode to measure the ratio of peak flow velocity in early versus late filling (E/A). In addition, the Tissue Doppler mode was used to assess peak septal mitral annulus velocity in the early filling phase (E′). For PW and Tissue Doppler mode, measurements from at least two cycles were averaged. The ratio of peak mitral flow versus annular velocity (E/E′) was calculated. The sphericity index was calculated as LVEDV/volume sphere, with volume sphere = (4*3.14/3)*((LVEDD/2)^3) and with LVEDD as the LV end-diastolic diameter. Analyses were performed using Vevo^®^ LAB (Vevo^®^ LAB software, version 5.6.1, FUJIFILM VisualSonics, Inc.). Two researchers analyzed all echocardiographic images.

#### 2.3.2. Strain Imaging

To perform strain analysis of the LV, mid-ventricular parasternal short- and long-axis B-mode cine loops were imported into the Vevo^®^ Strain software (version 5.6.1, FUJIFILM VisualSonics, Inc.), and the most optimal cardiac cycles were selected to assess LV global circumferential strain (LVGCS, short-axis), LV peak radial strain (long-axis), and LV global longitudinal strain (LVGLS, long-axis). The endocardial border was manually traced, and simultaneous epicardial tracing was performed automatically to obtain 48 sampling points, dividing the LV into six myocardial segments. In each segment, peak strain was analyzed.

Random missing values of echocardiographic parameters are due to poor image quality or deceased animals.

### 2.4. Hemodynamic Measurements

Invasive LV and right ventricle (RV) hemodynamic measurements were performed under 2% isoflurane anesthesia supplemented with oxygen, as described previously [31]. A pre-calibrated SPR-320 Mikro-Tip rat pressure catheter (Millar Inc., Houston, TX, USA) was placed in the LV via the right carotid artery or RV via the right external jugular vein. The catheter was connected to a quad-bridge amplifier and PowerLab 4/25T (AD Instruments, Oxford, UK). Functional LV and RV parameters, including end-systolic pressure (LVESP and RVESP), end-diastolic pressure (LVEDP and RVEDP), maximum and minimum peak time derivatives (LV dP/dt_max_ and RV dP/dt_max_, and LV dP/dt_min_ and RV dP/dt_min_, respectively), contractility index, and time constant for isovolumetric relaxation (LV tau and RV tau), were measured, ensuring stable and reliable recordings. Data were obtained and analyzed with LabChart 8 software (AD instruments).

### 2.5. Quantification of Plasma BNP Levels

Blood samples were centrifuged at 2000 rpm for ten minutes at 4 °C. Plasma was collected and stored at −20 °C until used to quantify brain natriuretic peptide (BNP). Plasma BNP was measured using the BNP 45 Rat ELISA Kit (ab108816, Abcam, Cambridge, UK) following the manufacturer’s guidelines. First, standards and plasma samples were added to the ELISA plate coated with a BNP-capturing antibody. Then, a biotinylated BNP 45 detector antibody was added, followed by a streptavidin-peroxidase conjugate and chromogen substrate solution. The unknown BNP concentrations were determined by comparison with the predetermined standard curves. Absorbance was measured at 450 and 570 nm using the Fluostar Optima plate reader (Clariostar Plus, BMG Labtech, Ortenberg, Germany). Readings at 570 nm were subtracted from those at 450 nm to correct optical imperfections.

### 2.6. Histology

Transverse sections of paraffin-embedded LV tissue (7 µm) were obtained at the level of the papillary muscles. These were stained with hematoxylin and eosin (H&E) to evaluate general tissue morphology, measure cardiomyocyte cross-sectional area (CSA) by assuming an elliptical cross-section using Fiji software (version 1.53t) [32], and assess the presence of leukocytes. Cardiomyocyte CSA was measured in at least 12 cells per animal. Semi-quantification of leukocytes was performed in the whole section, with score 1: limited number/absence, score 2: minor number, score 3: moderate number, score 4: high number, score 5: high number, and presence of aggregates. In addition, sections were stained using the Sirius Red/Fast Green Collagen Staining Kit to detect collagen deposition and assess interstitial and perivascular fibrosis (9046, Chondrex, Inc., Woodinville, WA, USA). After staining, sections were mounted using DPX. Imaging was performed with a Leica MC170 camera connected to a Leica DM2000 LED microscope (Leica Biosystems, Diegem, Belgium). The area of collagen deposition indicated by red staining was quantified in at least six randomly chosen and blinded images per section with the color deconvolution plugin in Fiji to determine the degree of interstitial fibrosis [32]. This area was normalized to the total area and expressed as % collagen deposition. Perivascular fibrosis was determined by dividing the fibrotic area, quantified using the color threshold tool, by the luminal area, quantified with the assumption of an elliptical cross-section, in Fiji [32]. Perivascular fibrosis was determined for at least three vessels located in the LV. All histological analyses were performed by two researchers independently.

### 2.7. Immunohistochemistry

Transverse sections of paraffin-embedded LV tissue (7 µm) were obtained at the level of the papillary muscles and stained for the oxidative stress marker nitrotyrosine, the macrophage marker CD68, or the marker of lipid peroxidation 4-hydroxynonenal (4-HNE). First, heat-mediated antigen retrieval was performed with citrate buffer (pH = 6). Endogenous peroxidase was blocked with 30% hydrogen peroxide (H_2_O_2_) diluted 1:100 in PBS for 20 min at room temperature. Tissue sections were permeabilized with 0.05% Triton X-100 (Merck Life Science B.V., Hoeilaart, Belgium) and blocked with serum-free protein block (X0909, Dako, Agilent Technologies, Machelen, Belgium) to limit background staining, both for 20 min at room temperature. The sections were incubated with a primary antibody for nitrotyrosine (1:100, 1 h at room temperature, mouse monoclonal HM.11, ab7048, Abcam), CD68 (1:100, overnight at 4 °C, mouse monoclonal MCA341R, Bio-Rad, Temse, Belgium), or 4-HNE (1:400, overnight at 4 °C, mouse monoclonal [HNEJ-2], ab48506, Abcam). A biotinylated secondary antibody (1:100, 30 min, anti-mouse, E0413, Dako) and streptavidin-HRP (1:400, 30 min at room temperature, P0397, Dako) were used for nitrotyrosine. EnVision™ + Dual Link System-HRP (30 min at room temperature, anti-rabbit/anti-mouse, K4061, Dako) was used for CD68 and 4-HNE. Negative controls were included in each staining, with no primary antibody applied to the sections. Sections were incubated with 3,3′-Diaminobenzidine (DAB) solution (Dako), counterstained with hematoxylin, and mounted with DPX mounting medium. Imaging was performed with a Leica MC170 camera connected to a Leica DM2000 LED microscope (Leica Biosystems). The nitrotyrosine deposition was measured in four to eight randomly chosen and blinded images per section using the color deconvolution plugin in Fiji [32]. This area was normalized to the total area and expressed as %. Semi-quantification of CD68^+^ cells was performed in the whole section as described for leukocytes above. All immunohistochemical analyses were performed by two researchers independently.

### 2.8. Assessment of Cardiomyocyte Organization and Mitochondrial Structure

The tissue of the LV was processed to examine cardiomyocyte organization with light microscopy (LM) and transmission electron microscopy (TEM). Tissue was fixed overnight with 2% glutaraldehyde in 0.05 M cacodylate buffer at 4 °C, post-fixed in 2% osmium tetroxide, and stained with 2% uranyl acetate in 10% acetone. Samples were dehydrated in a graded acetone series and embedded in araldite using the pop-off method. Semi-thin sections were stained with toluidine blue. The mitochondrial area was assessed with LM and quantified using the threshold tool in Fiji [32]. The data were expressed as the percentage area occupied by mitochondria relative to the total cell area. In addition, ultra-thin sections were cut and mounted on formvar-coated grids, counterstained with uranyl acetate and lead citrate, and imaged in a Philips EM 208 transmission electron microscope (Eindhoven, The Netherlands). Per animal, five random TEM pictures of cardiomyocytes were taken at 1200× and 6000× magnification to perform a deep morphological analysis of intracellular cardiomyocyte organization. Mitochondrial density was calculated as the number of mitochondria, measured using the cell counter tool in Fiji [32], normalized to the total cell area. The mitochondrial, myofilament, and cytoplasmic surface areas were measured by grid-point analysis in Fiji [32], counting every point (distance = 2 μm) at the intersection of horizontal and vertical lines. Data were presented as a percentage of points hitting the different structures relative to total grid points. All samples were coded, and analyses were performed by two blinded researchers independently.

### 2.9. Real-Time PCR

Total RNA was extracted from LV tissue using the RNeasy Fibrous Tissue Kit (Qiagen Benelux B.V., Antwerp, Belgium) as described before and following the manufacturer’s guidelines [33]. The concentration and purity of RNA were assessed with the NanoDrop 200 spectrophotometer (Isogen Life Science B.V., Utrecht, The Netherlands). RNA was reverse-transcribed to cDNA using the qScript cDNA SuperMix (Quantabio, VWR International bvba, Leuven, Belgium). Primers were designed in the mRNA coding sequence (Appendix A). Real-time PCR was carried out in a MicroAmp™ Fast Optical 96-well reaction plate using the QuantStudio™ 3 Real-Time PCR System (Thermo Fisher Scientific, Geel, Belgium). SYBR Green (Thermo Fisher Scientific) chemistry-based qPCR was performed [34]. MIQE guidelines were considered when analyzing gene expression data [35]. The most stable reference genes for this experimental set-up (TATA box-binding protein (tbp) and ribosomal protein L13A (Rpl13a)) were determined by geNorm.

### 2.10. Cluster Analysis

The Ct values from qPCR expression data were log-transformed and Pareto-scaled to visualize the hierarchically clustered heatmaps of different sample groups. The heatmap.2 function in the gplots package was implemented in the R program (R 4.0.3) and was used to construct the heatmap.

### 2.11. Statistical Analysis

Statistical analysis was performed using GraphPad Prism (GraphPad software, version 9.5.0). The normal distribution of data were verified with the D’Agostino and Pearson normality test or the Shapiro–Wilk test when N < 8. For parameters measured at a single time point, the data were compared with the parametric one-way ANOVA test followed by the Bonferroni post hoc test or the non-parametric Kruskal–Wallis test followed by Dunn’s post hoc test. In cases of unequal variances, the Brown–Forsythe ANOVA test, followed by Dunnett’s T3 post hoc test, was used. For parameters measured at multiple time points, the data were compared with repeated measures two-way ANOVA or mixed-effects analysis followed by the Bonferroni post hoc test. The between-group difference in body weight over time was compared with the repeated measures one-way ANOVA test, followed by Bonferroni correction. The log-rank test was used for survival analysis. Outliers were identified using the ROUT method with a maximum desired False Discovery Rate of 1%, with this criterion established a priori. Data are expressed as the mean ± SEM; *p* < 0.05 was considered statistically significant.

## 3. Results

### 3.1. PM Protects against DOX-Induced LV Cardiomyopathy

DOX-treated animals gained significantly less weight from baseline to week 8 compared to CTRL (Appendix A). Some mortality occurred in the DOX group, but survival data were not significantly different between the DOX, DOX+PM, and CTRL groups (Appendix A). At baseline, no differences in echocardiographic parameters between groups were detected (Appendix A). Cardiac parameters at week 8 are shown in Table 1. DOX administration significantly reduced LVEF, longitudinal LV fractional shortening (LVFS), and radial LVFS. LV end-systolic (LVESV) and end-diastolic volume (LVEDV) were significantly increased after DOX, as well as sphericity index (*p* < 0.0001 for LVEF, LVFS, and LVESV, *p* = 0.0002 for LVEDV, and *p* = 0.0039 for sphericity index). LV anterior wall thickness in diastole (LVAW_d_), posterior wall thickness in diastole (LVPW_d_), and diastolic function (E/A and E/E′) were unchanged after DOX. DOX also significantly reduced non-conventional echocardiographic parameters, such as LV global longitudinal strain (LVGLS), LV global circumferential strain (LVGCS), and LV peak radial strain (all *p* < 0.0001). Plasma BNP was not increased after DOX. Overall, cardiac function was significantly improved in the DOX+PM group. Indeed, DOX+PM showed improved LVEF (+17% relative change, *p* = 0.0140), radial LVFS (+23% relative change, *p* = 0.0002), sphericity index (−29% relative change, *p* = 0.0264), LVGLS (+25% relative change, *p* = 0.0095), and LVGCS (+19% relative change, *p* = 0.0368) compared to DOX alone. LVEF, radial LVFS, and LVGCS remained different from the CTRL+PM group (*p* < 0.0001, *p* = 0.0004, and *p* = 0.0022, respectively), indicating partial cardioprotection by PM. Representative echocardiographic pictures of M-mode, 4D-mode, and strain for each group are shown in Appendix A.

Hemodynamic parameters of LV function assessed at sacrifice are shown in Table 2. DOX significantly reduced LV end-systolic pressure (LVESP, *p* < 0.0001), but LV end-diastolic pressure (LVEDP) was similar between the CTRL and DOX groups. In addition, a significant reduction of LV maximum peak time derivative (dP/dt_max_), LV contractility index, and LV minimum peak time derivative (dP/dt_min_) was observed in the DOX group (*p* < 0.0001 for dP/dt_max_ and dP/dt_min_, and *p* = 0.0193 for contractility index). Similarly to the echocardiographic parameters, PM prevented the deterioration of hemodynamic parameters. Indeed, the DOX+PM group showed higher LVESP, LV dP/dt_max_, and LV dP/dt_min_ compared to DOX.

RV function, assessed by examining RV hemodynamics, was unaffected by DOX (Appendix A), in line with the unchanged lung wet-to-dry weight ratio for all groups (Table 2).

### 3.2. PM Reduces DOX-Induced LV Fibrotic Remodeling

Typical LV sections stained with H&E and related analyses are shown in Figure 1a. The cardiomyocyte cross-sectional area was similar for all groups, in line with the unchanged LV wall thickness observed with echocardiography. The expression of the pro-apoptotic genes p53 and BAX was significantly higher in the DOX group (Figure 1b). With PM, expression levels were comparable to CTRL levels. Regarding LV tissue remodeling, DOX significantly increased both interstitial and perivascular collagen deposition (Figure 1c–e). Concomitant PM treatment significantly reduced collagen deposition at interstitial and perivascular sites when compared to DOX alone. Confirming these data, PM treatment could reduce the increased gene expression of type I collagen, type III collagen, LOX and TGF-β1 caused by DOX (Figure 1f).

### 3.3. PM Alleviates DOX-Induced Macrophage-Driven Myocardial Inflammation

DOX administration resulted in an increase in CD68+ cells, a marker of monocytes and macrophages, in LV tissue (Figure 2a–c). This finding was confirmed by increased CD68 gene expression in DOX-treated animals (Figure 2d, left panel). The expression was significantly lower in the DOX+PM group. DOX also induced a higher gene expression of the pro-inflammatory cytokines IL-6 and IL-1β, which was reduced with PM (Figure 2d, middle and right panels). Different macrophage subsets were also evaluated. After eight weeks of DOX administration, no difference was observed in the gene expression of CD86, a marker of antigen-presenting cells, compared to CTRL (Appendix A), while the gene expression of the anti-inflammatory macrophage marker CD163 was significantly increased with DOX and lowered by PM (Appendix A). The gene expression of CD206 remained similar in all groups (Appendix A).

### 3.4. PM Restores DOX-Induced Disturbance of Redox and Iron Regulation

DOX administration increased nitrotyrosine deposition in LV tissue (Figure 3a). In addition, quantification of 4-HNE protein levels revealed that lipid peroxidation was significantly increased in DOX-treated animals compared to CTRL, which could not be alleviated by PM (Figure 3b). The gene expression of pro-oxidative and antioxidative transcription factors, respectively, NF-κB and NRF-2, was significantly increased by DOX, while PM could restore these levels (Figure 3c). The gene expression of the antioxidative genes NQO1 and SOD2 was significantly lower in the DOX-treated animals compared to CTRL (Figure 3d). PM treatment increased NQO1 gene expression but did not affect SOD2 gene expression. DOX alone did not alter the expression of the gene TOP2β. However, the administration of PM significantly lowered TOP2β expression (Appendix A). To further unravel the underlying mechanisms responsible for the protective nature of PM, we examined the expression of genes involved in cellular iron regulation. These data revealed that DOX administration significantly increased the gene expression of the cytosolic iron importer ZIP14, iron storage gene FTH1, and mitochondrial iron importer MFRN-1, while the expression of the cytosolic iron exporter FPN was unchanged (Figure 3e). Concomitant PM treatment significantly lowered the gene expression of ZIP14, FTH1, and MFRN-1. The expression of other genes involved in iron metabolism was unchanged with DOX (Appendix A), except for NRAMP2, responsible for the transport of iron from endosomes to the cytosol, which was significantly reduced after PM treatment (Appendix A).

### 3.5. PM Attenuates DOX-Induced Mitochondrial Damage

As shown in Figure 4a, DOX administration resulted in an evident distortion of cardiomyocyte organization. This was attenuated by PM as well as the DOX-induced reduction of mitochondrial area (Figure 4b). Ultrastructural examination of cardiomyocytes revealed mitochondrial damage with swollen mitochondria, disrupted cristae, and autophagic vacuoles after DOX administration (Figure 4c). There is a trend toward lower mitochondrial density in the DOX group (Figure 4d, left panel), while this was significantly increased by PM. Grid-point analysis showed no differences in the mitochondrial and myofilament fractions, while the cytoplasmic fraction was significantly increased with DOX (Appendix A).

### 3.6. PM Improves Adverse Gene Expression Patterns Involved in DOX-Induced Cardiotoxicity

The gene expression analysis of 24 genes involved in inflammation, oxidative stress, apoptosis, and fibrosis revealed a highly distinguished profile between the DOX and CTRL groups, suggesting that DOX induces profound changes in gene expression (Figure 5a,b). The redox regulation was disturbed in the DOX-treated group, as shown by the downregulation or upregulation of (anti)oxidative stress-related genes (HO-1, NOX2, NOX4, NQO1, NRF-2, SOD1, and CAT), whereas PM restored the expression towards the CTRL profile. In addition, PM downregulated the relative gene expression of apoptosis-related genes (p53, BAX, and Bcl-2), inflammatory-associated genes (NF-κB and IL-6), and fibrosis genes (LOX, type I collagen, and type III collagen) compared with the DOX-treated group. Overall, DOX treatment shifted the gene expression profile of many genes related to inflammation, apoptosis, fibrosis, and oxidative stress, whereas PM partially prevented this shift and moved toward the CTRL profile.

## 4. Discussion

### 4.1. PM as a Cardioprotective Strategy during DOX Treatment

Basic research should be focused on identifying new cardioprotective therapies that allow the optimal use of cancer treatments while minimizing cardiotoxicity. This study is the first to report on the potential cardioprotective effect of PM in an animal model of DOX-induced cardiomyopathy. The protective mechanisms of PM are multifactorial, including reduced fibrotic remodeling and myocardial inflammation, restored redox and iron regulation, and limited mitochondrial damage and ferroptosis. In our study, we performed state-of-the-art techniques, including 4D echocardiography and myocardial strain analysis, to assess LV cardiac function, in line with the strategy proposed by Čelutkienė et al. (2018) [36] and Zacchigna et al. (2021) [30], together with hemodynamic measurements. After eight weeks of DOX administration, we observed typical DOX-induced remodeling matching LV dilated cardiomyopathy and systolic dysfunction, characterized by increased LV volumes, reduced LVEF and LVFS, altered strain parameters (i.e., LVGLS, LVGCS, and LV peak radial strain), and disturbed LV hemodynamics. Together with the fact that DOX was administered intravenously for eight weeks, which is the relevant administration route, these data indicate that our DOX-induced cardiomyopathy model are valid and translatable. DOX-induced cardiomyopathy is typically characterized by systolic dysfunction, although some studies state that diastolic dysfunction is one of the hallmarks of DOX-induced cardiomyopathy, often preceding systolic dysfunction [37]. We did not observe diastolic dysfunction based on echocardiographic evaluation (i.e., E/A and E/E′), which can be explained by the fact that hemodynamic measurements are less influenced by changes in cardiac load and more accurately reflect myocardial contractility in rodents. Next to systolic dysfunction and thus forward failure, we also investigated whether DOX resulted in backward failure and lung congestion. Hemodynamic measurements revealed that RV function was unchanged, as also shown by others [38], and the wet-to-dry lung weight ratio was similar between groups, indicating the absence of congestion.

In our animal model of DOX-induced cardiomyopathy, we show that the eight-week co-administration of PM, a natural form of vitamin B6 known for its capacity to chelate metal ions and its anti-inflammatory and antioxidant properties [39], protects against the development of DOX-induced LV cardiomyopathy and systolic dysfunction without inducing side effects. Rats from the DOX+PM group showed improved LVEF, LVFS, and LVGLS and reduced LV volumes and fibrotic remodeling compared to DOX alone. In addition, PM treatment prevented DOX-induced LV hemodynamic disturbances with improved LVESP, LV dP/dt_max_, and LV dP/dt_min_. Previously, we have shown that 1 g/L PM intake is safe and effectively limits LV deformation and fibrosis development by reducing type I and III collagen levels, the major collagen types of the myocardial extracellular matrix, in rats with myocardial infarction [20] and glycated protein-induced adverse vascular [31] and cardiac remodeling [33]. In this study, we demonstrate that PM has similar effects in a DOX-induced cardiotoxicity model as it prevents the DOX-induced increase in interstitial and perivascular myocardial fibrosis by reducing the expression of type I and III collagen levels. We also show that the beneficial effect of PM on myocardial fibrosis is mainly due to a decreased stimulation of the TGF-β1 signaling pathway, an important regulator of fibrosis development [40]. These findings are confirmed by El-Agamy et al. (2019), who have observed increased type I collagen and TGF-β protein expression in the cardiac tissue of DOX-treated animals [41]. Furthermore, in addition to reducing the amount of collagen deposition, PM treatment also lowered LOX expression, consequently limiting excessive collagen cross-linking and myocardial wall stiffening induced by DOX. This inhibition of myocardial fibrosis and wall stiffening is reflected in improved cardiac function and the prevention of dilated cardiomyopathy, as shown by echocardiography and hemodynamic measurements.

### 4.2. Underlying Mechanisms of PM Cardioprotection

Inflammation and oxidative stress play a central role in cardiac damage caused by DOX [7]. We observed immune cell infiltration in the LV tissue of DOX-treated rats, with a significant increase in CD68^+^ macrophages and higher gene expression levels of pro-inflammatory cytokines IL-6 and IL-1β in the LV tissue. PM reduced the gene expression of IL-6 and CD68, illustrating its anti-inflammatory nature. Zhang et al. (2020) also observed DOX-induced macrophage accumulation and overexpression of IL-6 and IL-1β in the cardiac tissue of DOX-treated animals, which was induced through the proliferation of resident macrophages by activating the scavenger receptor class A1-c-Myc axis [42]. In addition, they demonstrated dynamic changes in macrophage subtypes that dominate during the process of DOX-induced myocardial inflammation. The pro-inflammatory macrophages dominate in the initial phase of DOX cardiomyopathy, indicating profound macrophage-induced myocardial inflammation, whereas reparative macrophages slowly increase along the disease progression, thereby eliciting a compensatory response. The dominant role of reparative macrophages in the later phase of DOX-induced myocardial inflammation supports our finding that, after eight weeks of DOX treatment, the expression of the anti-inflammatory macrophage marker CD163 but not CD86 was significantly increased compared to CTRL. DOX administration was also associated with the induction of NF-κB signaling, which was attenuated by PM treatment. This finding explains the observed increase in IL-6 expression by DOX because NF-κB functions as a transcription factor for pro-inflammatory (and pro-oxidative) genes, including IL-6, involved in inflammatory reactions and cardiac damage, which is in line with El-Agamy et al. (2019) [41].

Mitochondria, abundantly present in cardiac tissue, are a clear subcellular target of DOX, and several cardioprotective strategies focus on targeting these organelles [43]. Our study shows that PM offers cardioprotection by preventing the development of typical DOX-induced mitochondrial damage characterized by mitochondrial swelling, cytoplasmic vacuolization, and chromatin condensation. Moreover, mitochondrial area and density were increased in the DOX+PM group. Previous studies have shown that DOX accumulates in mitochondria due to its high affinity for cardiolipin, a phospholipid abundant in the inner mitochondrial membrane, disturbing mitochondrial function and metabolism via different mechanisms [43]. For example, DOX disrupts the mitochondrial respiratory chain, resulting in the formation of ROS, which may trigger compensatory changes, including activation of the Keap1/NRF-2 pathway. NRF-2 is a transcription factor that regulates the expression of antioxidant proteins, and its activation is repressed by Keap1. In response to DOX-induced oxidative cell stress, the Keap1-NRF-2 complex dissociates, increasing free NRF-2 [7]. Indeed, we show increased NRF-2 gene expression in LV tissue, which was also observed by Li et al. (2021) in rat cardiomyoblasts [44], and we observed that PM treatment significantly lowered NRF-2 gene expression. In addition, PM partially restored NQO1 expression, a factor that prevents quinone ring reduction to a semiquinone radical (i.e., redox cycling) [45], further confirming its antioxidant properties. However, genetic analyses revealed that low NQO1 activity was not associated with increased heart failure risk in childhood cancer survivors treated with anthracyclines, questioning the clinical benefit of increasing NQO1 levels [46].

In the process of DOX redox cycling and DOX-induced cardiotoxicity in general, iron plays a pivotal role [9]. In the presence of iron, and particularly iron overload, H_2_O_2_ formed during redox cycling is converted into •OH radicals via the Fenton reaction. These radicals react with membrane lipids and induce lipid peroxidation, generating lipid radicals like 4-HNE [47]. Multiple studies support the detrimental role of iron overload in DOX-induced cardiotoxicity [11]. Patients with DOX-induced cardiomyopathy display higher iron levels in heart tissue than those without cardiomyopathy [9]. In addition, rats fed an iron-rich diet show more severe cardiomyocyte damage than rats fed a standard diet [48]. In our study, we show an increased gene expression of the iron importer ZIP14 and iron storage gene FTH1 in the DOX group, which is also reported by other in vitro and in vivo studies [44,49]. In combination with the unchanged gene expression of the iron exporter FPN, we can suggest that DOX disrupts cardiac iron regulation at the gene level by increasing cytosolic iron import rather than altering iron export. Disruption of cytosolic iron regulation likely results in mitochondrial iron disturbances. Indeed, Ichikawa et al. (2014) showed that iron accumulates specifically in the mitochondria due to DOX-induced suppression of ABCB8, resulting in reduced iron export, and that reversal of mitochondrial iron accumulation mitigates DOX-induced cardiotoxicity in mice [9]. In our study, mitochondrial iron overload is due to increased MFRN-1-mediated iron import in mitochondria at the gene level rather than reduced export since ABCB8 gene expression was unaltered.

Iron overload triggers a specific form of iron-dependent cell death called ferroptosis [8]. Fang et al. (2019) and Tadokoro et al. (2020) identified mitochondria-dependent ferroptosis as an important mechanism underlying DOX-induced cardiomyopathy and an interesting target for cardioprotection via iron chelation [50,51]. In addition to (mitochondrial) iron overload, DOX-induced ferroptosis is also characterized by lipid peroxidation, the generation of lipid radicals such as 4-HNE, and mitochondrial ultrastructural abnormalities including swelling, loss of cristae, and smaller mitochondria [8,52]. These hallmarks (i.e., lipid peroxidation and mitochondrial abnormalities) and increased NRF-2 expression—a genetic hallmark of ferroptosis [52]—are all confirmed in our study; which clearly show the presence of ferroptosis in our model of DOX-induced cardiomyopathy. In contrast, we could not confirm the downregulation of glutathione peroxidase 4 by DOX, which is a key feature of ferroptosis [51]. Interestingly, we show that PM treatment partially protects against DOX-induced ferroptosis, as evidenced by restored iron regulation and reduced mitochondrial abnormalities at the gene level, highlighting its promising cardioprotective nature.

### 4.3. Current Limitations and Future Perspectives

Taken together, concomitant PM treatment partially protects against DOX-induced LV cardiomyopathy by reducing LV fibrotic remodeling, myocardial inflammation, the disturbance of redox and iron regulation, and mitochondrial damage. However, we recognize the limitation that these pathways were only examined at the transcript level. Future experiments should confirm these findings on the protein level. As a result, we are the first to discover PM as a potential multifactorial cardioprotective strategy for DOX-induced cardiomyopathy in a translatable animal model. In clinical trials with dialysis patients and patients with diabetic nephropathy, PM administration was safe and showed beneficial outcomes [53,54]. However, the cardioprotective effect of PM remains to be tested in DOX-treated cancer patients.

The main limitation of this study is the use of healthy animals without tumors. However, this is the case for most published studies [24] and will be the subject of our further research. In our lab, we have already observed that PM does not interfere with the potential of DOX to reduce proliferation and viability and to increase cytotoxicity and apoptosis of LA7 rat mammary tumor cells in vitro [55].

## Figures and Tables

**Figure 1 antioxidants-13-00112-f001:**
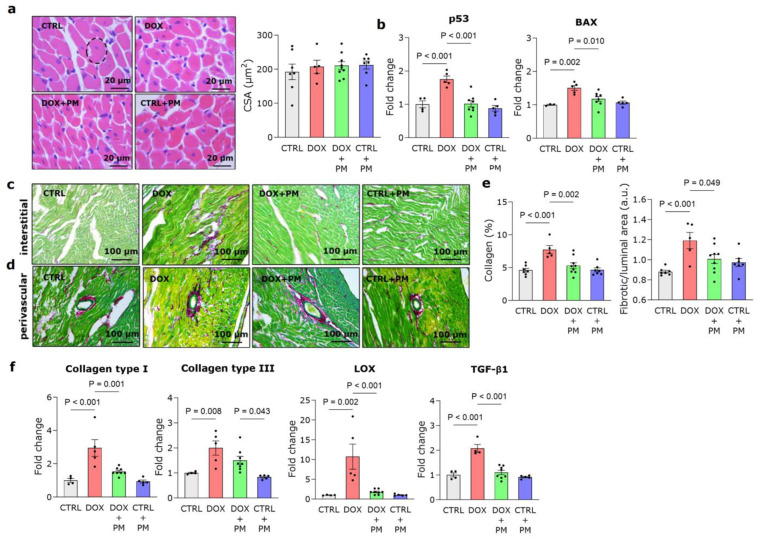
PM reduces DOX-induced fibrotic remodeling. (**a**) Representative images of LV tissue stained with H&E and quantification of cross-sectional cardiomyocyte area (CSA) in CTRL (N = 7), DOX (N = 5), DOX+PM (N = 9), and CTRL+PM (N = 7) animals. The dashed line represents an elliptical representation of one cardiomyocyte. (**b**) Quantification of the expression of pro-apoptotic genes p53 and BAX in LV tissue from CTRL (N = 3–4), DOX (N = 5), DOX+PM (N = 8), and CTRL+PM (N = 5) animals. (**c**–**e**) Representative images of LV tissue stained with Sirius Red/Fast Green (**c**,**d**) and quantification of the percentage of interstitial and perivascular collagen deposition (**e**) in CTRL (N = 7), DOX (N = 5), DOX+PM (N = 9), and CTRL+PM (N = 7) animals. (**f**) Quantification of the expression of type I collagen, type III collagen, LOX, and TGF-β1 in LV tissue from CTRL (N = 4), DOX (N = 4–5), DOX+PM (N = 8), and CTRL+PM (N = 5) animals. The data are shown as mean ± SEM. DOX, doxorubicin. LV, left ventricular. PM, pyridoxamine.

**Figure 2 antioxidants-13-00112-f002:**
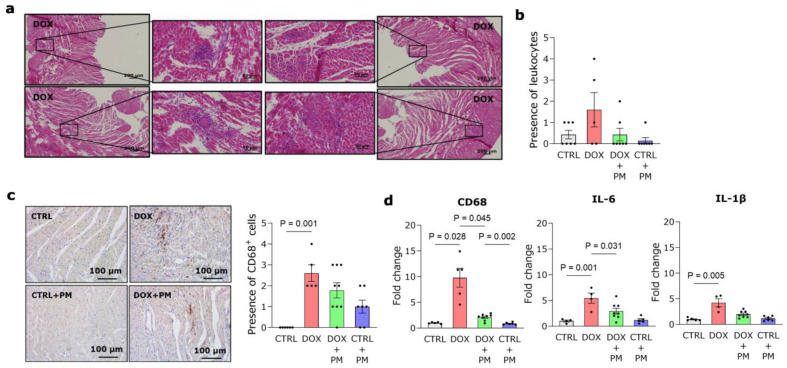
PM reduces DOX-induced myocardial inflammation. (**a**,**b**) Representative images of LV tissue from DOX-treated rats stained with H&E (**a**) and semi-quantification of leukocytes in LV tissue from CTRL (N = 7), DOX (N = 5), DOX+PM (N = 7), and CTRL+PM (N = 7) animals (**b**). (**c**) Representative images and semi-quantification of CD68+ cells in LV tissue sections from CTRL (N = 6), DOX (N = 5), DOX+PM (N = 9), and CTRL+PM (N = 7) animals. (**d**) Quantification of CD68, IL-6, and IL-1β gene expression in LV tissue from CTRL (N = 4–5), DOX (N = 4–5), DOX+PM (N = 8), and CTRL+PM (N = 5–6) animals. The data are shown as mean ± SEM. DOX, doxorubicin. LV, left ventricular. PM, pyridoxamine.

**Figure 3 antioxidants-13-00112-f003:**
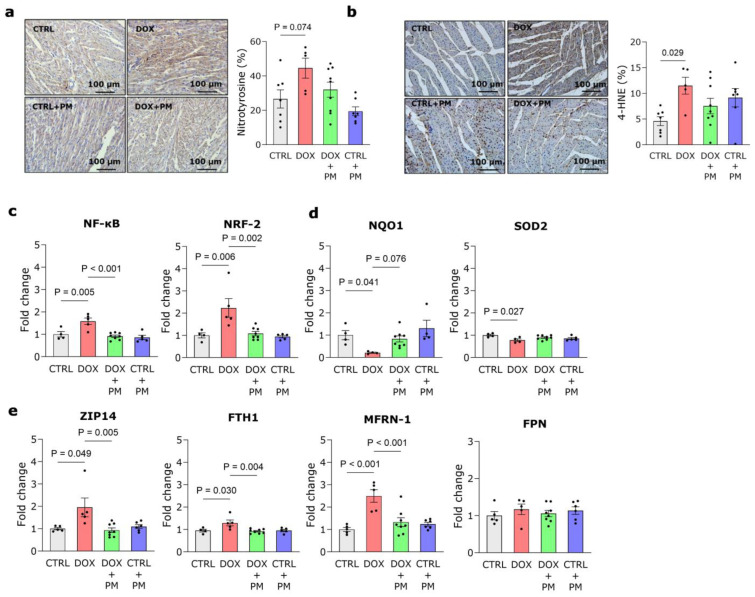
PM restores DOX-induced disturbances of redox and iron regulation. (**a**,**b**) Representative images and quantification of nitrotyrosine staining (**a**) and 4-hydroxynonenal (4-HNE) staining (**b**) in LV tissue from CTRL (N = 7), DOX (N = 5), DOX+PM (N = 9), and CTRL+PM (N = 7) animals. (**c**–**e**) Quantification of the expression of genes involved in redox regulation (NF-κB, NRF-2, NQO1, and SOD2) (**c**,**d**) and iron handling (ZIP14, FTH1, MFRN-1, and FPN) (**e**) in LV tissue from CTRL (N = 4–5), DOX (N = 4–5), DOX+PM (N = 8), and CTRL+PM (N = 4–6) animals. The data are shown as mean ± SEM. DOX, doxorubicin. LV, left ventricular. PM, pyridoxamine.

**Figure 4 antioxidants-13-00112-f004:**
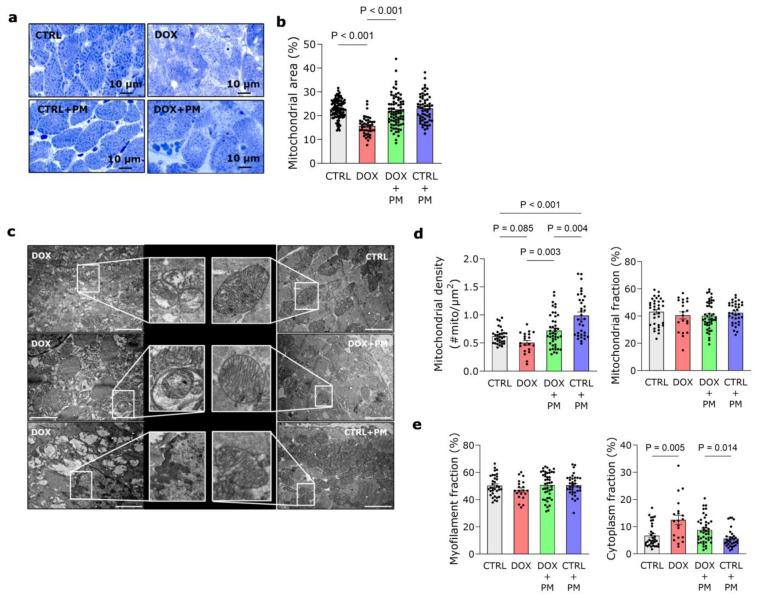
PM attenuates DOX-induced mitochondrial damage. (**a**) Representative images of toluidine blue-stained LV tissue showing clear distortion of cardiomyocyte organization in the DOX group. (**b**) Quantification of mitochondrial area relative to total cell area in CTRL (ncells = 108), DOX (ncells = 44), DOX+PM (ncells = 77), and CTRL+PM (ncells = 60) cardiomyocytes. (**c**) Representative transmission electron microscopy (TEM) pictures of LV cardiomyocytes. In the DOX group, cardiomyocytes showed swollen mitochondria, disrupted cristae, autophagic vacuoles containing damaged mitochondria, cytoplasmic vacuoles, and condensed chromatin, which was less pronounced in the DOX+PM group. Cardiomyocytes of CTRL and CTRL+PM groups showed typical intracellular organization of myofilaments (light grey), mitochondria (dark grey), and cytoplasm (white). Magnification: 6000×. Scale bars: 2 μm. (**d**) Quantification of percentage mitochondrial density, calculated as the mitochondrial number relative to the total cell area, and mitochondrial fraction, expressed as a percentage of total grid points in CTRL (ncells = 32–34), DOX (ncells = 20), DOX+PM (ncells = 44), and CTRL+PM (ncells = 34–35) cardiomyocytes. (**e**) Quantification of myofilament and cytoplasm fraction, expressed as a percentage of total grid points, in CTRL (ncells = 34), DOX (ncells = 20), DOX+PM (ncells = 43–44), and CTRL+PM (ncells = 33–35) cardiomyocytes. For (**b**,**d**,**e**): NCTRL = 7, NDOX = 4, NDOX+PM = 9, NCTRL+PM = 5–7. The data are shown as mean ± SEM. DOX, doxorubicin. LV, left ventricular. PM, pyridoxamine.

**Figure 5 antioxidants-13-00112-f005:**
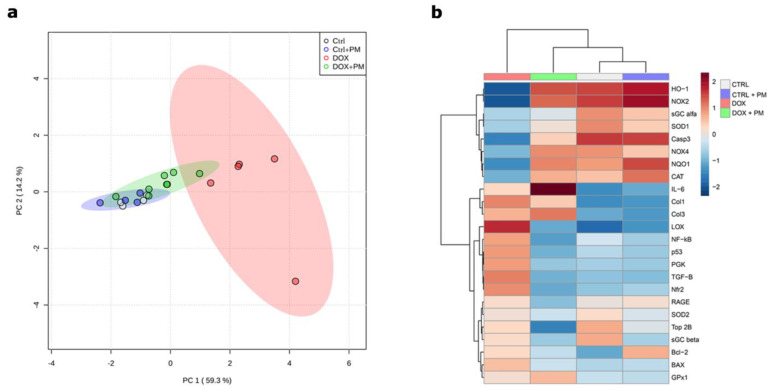
PM partially prevents the DOX-induced shift of genes related to inflammation, apoptosis, fibrosis, and oxidative stress. (**a**) PCA score plots of the first two principal components (PC) for qPCR gene expression of LV tissue samples from CTRL (N = 4), DOX (N = 4–5), DOX+PM (N = 8), and CTRL+PM (N = 4–5) animals. (**b**) Heatmap and hierarchical clustering show differentially expressed genes in LV tissue of the four experimental groups. Each column contains the average gene expression in LV tissue samples from CTRL (N = 4), DOX (N = 4–5), DOX+PM (N = 8), and CTRL+PM (N = 4–5) animals. Relative gene expression is indicated by color code: up-regulation (red) and down-regulation (blue). DOX, doxorubicin. LV, left ventricular. PCA, principle components analysis. PM, pyridoxamine.

**Table 1 antioxidants-13-00112-t001:** LV Echocardiographic parameters and plasma BNP at week 8.

	CTRL	DOX	DOX+PM	CTRL+PM
Conventional echocardiography				
LVEF (%)	86.0 ± 1.1	61.0 ± 3.3 ****	73.5 ± 1.9 ^†,‡‡‡‡^	84.6 ± 1.0
Longitudinal LVFS (%)	26.9 ± 0.8	14.7 ± 0.9 ****	19.1± 0.8 ^‡‡^	25.2 ± 1
Radial LVFS (%)	56.0 ± 1.4	33.3 ± 2.2 ****	43.4 ± 1.5 ^†††,‡‡‡^	53.0 ± 1.5
LV cardiac index (mL/min/m^2^)	0.14 ± 0.01	0.14 ± 0.01	0.15 ± 0.01	0.15 ± 0.01
LVESV/BSA (µL/cm^2^)	0.059 ± 0.004	0.300 ± 0.035 ****	0.166 ± 0.014 ^‡‡^	0.080 ± 0.007
LVEDV/BSA (µL/cm^2^)	0.504 ± 0.015	0.746 ± 0.041 ***	0.620 ± 0.026 ^‡^	0.517 ± 0.020
LVSV index (µL/cm^2^)	0.43 ± 0.01	0.45 ± 0.02	0.45 ± 0.02	0.44 ± 0.01
LV sphericity index	0.18 ± 0.01	0.27 ± 0.03 **	0.19 ± 0.01 ^†^	0.17 ± 0.01
LVAW_d_ (mm)	1.96 ± 0.05	1.85 ± 0.10	1.90 ± 0.06	1.77 ± 0.05
LVPW_d_ (mm)	1.86 ± 0.05	1.74 ± 0.09	1.73 ± 0.05	1.84 ± 0.05
E/A	1.41 ± 0.08	1.43 ± 0.08	1.40 ± 0.05	1.38 ± 0.05
E/E′	−25.2 ± 1.6	−28.1 ± 2.3	−27.2 ± 1.3	−26.4 ± 2.0
HR (bpm)	341.3 ± 11.3	307.2 ± 9.8	339.5 ± 5.2 ^†^	330.1 ± 7.1
BSA (cm^2^)	455.5 ± 3.3	403.7 ± 9.0 ***	430.8 ± 5.2	447.9 ± 3.9
Strain				
LVGLS (%)	−37.7 ± 1.8	−24.0 ± 2.3 ****	−31.9 ± 1.5 ^††^	−35.9 ± 1.7
LVGCS (%)	−39.6 ± 1.7	−25.4 ± 2.2 ****	−31.2 ± 1.1 ^†,‡‡^	−39.1 ± 1.2
LV peak radial strain (%)	92.2 ± 7.2	47.1 ± 5.2 ****	58.7 ± 4.7 ^‡‡^	86.9 ± 7.6
Plasma				
BNP (ng/mL)	0.16 ± 0.03	0.24 ± 0.04	0.18 ± 0.02	0.12 ± 0.01

LV echocardiographic parameters and plasma BNP were measured in CTRL (N = 14), DOX (N = 14), DOX+PM (N = 18), and CTRL+PM (N = 14) animals after eight weeks of DOX injections and PM treatment. Random missing values are due to poor image quality. For plasma BNP: CTRL (N = 12), DOX (N = 11), DOX+PM (N = 18), and CTRL+PM (N = 10). Data are presented as mean ± SEM. ** *p* < 0.01, *** *p* < 0.001, and **** *p* < 0.0001 vs. CTRL. ^†^ *p* < 0.05, ^††^ *p* < 0.01, and ^†††^ *p* < 0.001 vs. DOX. ^‡^ *p* < 0.05, ^‡‡^ *p* < 0.01, ^‡‡‡^ *p* < 0.001, and ^‡‡‡‡^ *p* < 0.0001 vs. CTRL+PM. A, mitral flow velocity in the late filling phase. BNP, brain natriuretic peptide. BSA, body surface area. DOX, doxorubicin. E, mitral flow velocity in the early filling phase. E′, peak septal mitral annulus velocity in the early filling phase. HR, heart rate. LV, left ventricular. LVAWd, LV anterior wall in diastole. LVEF, LV ejection fraction. LVEDV, LV end-diastolic volume. LVESV, LV end-systolic volume. LVFS, or LV fractional shortening. LVGCS, LV global circumferential strain. LVGLS, LV global longitudinal strain. LVPWd, LV posterior wall in diastole. LVSV, LV stroke volume. PM, pyridoxamine.

**Table 2 antioxidants-13-00112-t002:** LV hemodynamic parameters at week 8.

	CTRL	DOX	DOX+PM	CTRL+PM
Hemodynamic parameters				
LVESP (mmHg)	99.8 ± 3.1	68.9 ± 5.4 ****	95.1 ± 1.2 ^††††^	95.3 ± 3.8
LVEDP (mmHg)	6.1 ± 1.6	8.6 ± 2.2	5.3 ± 0.7	7.3 ± 1.7
LV dP/dt_max_ (mmHg/s)	6750.0 ± 347.2	3322.0 ± 644.7 ****	5718.0 ± 287.3 ^††^	5861.0 ± 405.1
LV dP/dt_min_ (mmHg/s)	−7895.0 ± 655.3	−3199.0 ± 568.0 ****	−7182.0 ± 283.2 ^†††^	−6795.0 ± 119.4
LV contractility index (1/s)	114.8 ± 5.1	88.5 ± 7.0 *	102.5 ± 2.4	114.1 ± 9.4
Organ weight				
Lung wet-to-dry weight ratio	5.03 ± 0.09	5.41 ± 0.16	5.27 ± 0.10	4.95 ± 0.09

LV hemodynamic parameters were measured in CTRL (N = 7), DOX (N = 4), DOX+PM (N = 9), and CTRL+PM (N = 5) animals at sacrifice. Data are presented as mean ± SEM. * *p* < 0.05; **** *p* < 0.0001 vs. CTRL; ^††^ *p* < 0.01, ^†††^ *p* < 0.001, and ^††††^ *p* < 0.0001 vs. DOX. DOX, doxorubicin. LV, left ventricular. LV dP/dtmax, LV maximum peak time derivative. LV dP/dtmin, LV minimum peak time derivative. LVEDP, LV end-diastolic pressure. LVESP, LV end-systolic pressure. LV tau, LV time constant for isovolumetric relaxation. PM, pyridoxamine.

## Data Availability

The data presented in this study are available from the corresponding author on request.

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
