# Peer review of "Pyridoxamine Limits Cardiac Dysfunction in a Rat Model of Doxorubicin-Induced Cardiotoxicity"

_antioxidants, 2024, doi:10.3390/antiox13010112_

Round 1

Reviewer 1 Report

Comments and Suggestions for Authors

In the current study the authors investigated the effect of PM on DOX-induced heart failure in an experimental rat model. The authors suggest that PM may get a cardioprotective strategy against DOX-induced HF.

Main comments:

Although the study seems to be performed quite carefully, it is nearly impossible to jedge about the experiments and the validation of the data.

There is a complete confusion between Table 1 and Table 2. Does Table 2 show LV as in line 317 and in the title mentioned or RV data as indexed in the Table?

Figure 1: Collagen 3 is missing (double presentation of collagen 1). Why are Collagen 1, 3, and LOX are part of Fig. 1F but TGFß1 separated as 1G? Why are the n numbers so different and how did the authors select animals used for this parameter?

Why appear p values in Figure 3 with p> 0.05?

Statistics, such as in Fig. 2D, require Bonferroni corrections.

What is the difference between heat map and Fig. 3 with respect to SOD2? In one case it is reduced in the other case not different?

Further points regarding the concept:

If fibrosis is so prominent why do the authors obtain systolic failure but not diastolic HF as expected by fibrosis?

Do the animals really have cardiomyopathy? They have a reduced LVEF but this is still in a normal range transferred to clinical data (far above 60%) and there is no increase in BNP. Although stated in a different way these rats are not really sick. Therefore, it is difficult to conclude that you can protect the heart. There is no benefit in hard HF criteria such as RVF and increased liver wet weight.

I miss any information about heart weights (LV and RV).

Reviewer 2 Report

Comments and Suggestions for Authors

This article reports on interesting study to investigates the potential cardioprotective effects of pyridoxamine(PM). Doxorubicin(DOX) chemotherapy is limited by dose-dependent cardiotoxicity. This study suggests that PM may serve as a promising therapeutic approach to mitigate the adverse cardiovascular effects associated with doxorubicin chemotherapy, including reducing myocardial fibrosis, inflammation, and mitochondrial damage, and by restoring redox and iron regulation. They found PM may be a novel cardioprotective strategy for DOX-induced cardiomyopathy. The topic is intriguing and the article presents results of potential interest; however, there are several weaknesses that need to be addressed.

1.     The Safety Concerns of PM can be discussed. Some journals showed high doses of PM may damage the kidneys. How about the dose used in the mice?

2.     The design uses healthy animals, is the background of using DOX drugs different from the real clinical situation? What are the key considerations in designing an animal model to investigate the cardioprotective effects of PM in the context of doxorubicin-induced cardiotoxicity, and how can these considerations enhance the translational relevance of the findings to human patients? Is the dosage and time of the drug administered to the animals consistent with the clinical situation? The study could involve dose-response experiments to determine the most effective and safe dosage of PM.

3.     What is the appropriate translation of the optimal dose of PM identified in preclinical models of DOX-induced cardiotoxicity to human dosing, and how does this compare to traditional cardioprotective treatments in terms of safety and efficacy?

4.     In fig3B, 4-HNE represents an experiment of lipid peroxidation, but the results found that the degree of lipid peroxidation in the group given normal PM was also higher. Lipid peroxidation is also one of the causes of cardiovascular disease. Does it mean that PM has an impact on lipid peroxidation?

5.     In fig5B, The expression levels of RAGE, SOD2, and Top2B in the normal group with PM protection were even the same as those in the group given DOX. Why is it and is it meaningful?

Reviewer 3 Report

Comments and Suggestions for Authors

In this study, the authors investigated the protective role of pyridoxamine (PM) in a rat model of doxorubicin (DOX)-induced cardiotoxicity. The authors focused on cardiac functional recovery, reduction in fibrosis, inflammation and mitochondrial damage in PM supplemented animals with DOX-induced cardiotoxicity. These observations are complemented by a restored expression of the genes involved in the redox and iron regulation in these PM supplemented rats.

The study is interesting and well conducted. The results are robust and the manuscript is well written and illustrated.

 Specific comments:

 - One of my main comments is that the analyses concerning the iron, redox and apoptosis pathways were carried out at the transcript level only. The conclusions in the abstract and discussion tend to be over-interpreted whereas the verifications at the protein and functional scales were not carried out. The abstract and discussion should be revised to incorporate this limitation, in particular lines 25-30, 531-533, 579-583, 587-589, 601-604. And this aspect should be added to the limitations.

 - Lines 463 to 477: This part of the discussion is really an introduction and should be used to introduce the study and support the authors' work.

 - Table 2: this table concerned LV as mentioned in legend and text, RV was copied by mistake

 - Figure 1: Collagen type I was presented 2 times instead of Collagen III

 - Lines 361 and 368: CD68+ is a marker of monocytes and macrophages, not of the immune cell infiltrate; on the other hand, CD86+ concerns dendritic cells, B cells etc... all antigen-presenting cells. This needs to be corrected.

 -Line 389: “PM treatment increased NQO1 gene expression but did not affect SOD2 gene expression”, Expression of the NQO1 gene is partially restored as it is similar to controls but not significantly different from DOX (p=0.076). This should also be reviewed in the discussion on lines 565-566.

 - Lines 415-416: “Mitochondrial density was lower in the DOX group (Figure 4d, left panel)”, p=0.085, it’s not significant!

 - Materials and Methods section is very long and can be reduced.

The end of “study design”, from line 100 to 106, can be shortened because most of the information is included in the dedicated paragraphs (immune, TEM, qPCR etc…)

Semi-quantification of CD68+ and leukocytes is the same, no need to repeat.

All the genes studied by qPCR are already listed in the Suppl. Table 1. There is no need to describe everything in the methods section from line 240 to line 254.

Perhaps some parts of the Materials and Methods can be placed into the supplemental materials?

 Minor comment:

 - For histology and immunology, how are the sections chosen? Was only one section analysed for each heart? Where were they taken: basal, median? In the same way, Figure 2, why 4 images from the 5 DOX group and none of the other groups?

 - Line 262: “The mRNA values from qPCR expression”, do you mean Ct value?

 - Line 324: “to DOX” is repeated

 -Line 334: “lung wet-to-dry weight ratio” concerns the Table 2, not the Suppl. Table 3

 -Line 398-399: “iron metabolism was unchanged with DOX (Figures 4b, c, left panel, and d)”, it’s in Figure S4b

Round 2

Reviewer 1 Report

Comments and Suggestions for Authors

I thank the authors for the corrections. The addition of the increased liver weights may argue to HF. I still be puzzled why the hearts did not have a diastolic defect as well. I understand your arguments why they develop systolic failure but difficult to assume that fibrosis does not lead to an increase diastolic stiffness. P values above 0.05 may be helpful but it might be better to replace by exact values to avoid confusion. Small effects may be functional relevant, I agree. This can be precised by giving effect sizes in future studies.

Reviewer 3 Report

Comments and Suggestions for Authors

Congratulations to the authors, who have considerably improved their manuscript and provided clear answers to the questions raised in the first report. The manuscript is of publishable quality.